ecology

community assembly, functional traits, latitudinal gradient, predation, historical contingency, eelgrass epifauna

**Author for correspondence:**
Collin P. Gross
e-mail: colgross@ucdavis.edu

# The biogeography of community assembly: latitude and predation drive variation in community trait distribution in a guild of epifaunal crustaceans

Collin P. Gross[1], J. Emmett Duffy[3], Kevin A. Hovel[4], Melissa R. Kardish[1], Pamela L. Reynolds[2], Christoffer Boström[5], Katharyn E. Boyer[6], Mathieu Cusson[7], Johan Eklöf[8], Aschwin H. Engelen[9], Britas Klemens Eriksson[10], F. Joel Fodrie[11], John N. Griffin[12], Clara M. Hereu[13], Masakazu Hori[14], A. Randall Hughes[15], Mikhail V. Ivanov[16], Pablo Jorgensen[17], Claudia Kruschel[18], Kun-Seop Lee[19], Jonathan Lefcheck[2], Karen McGlathery[20], Per-Olav Moksnes[21], Masahiro Nakaoka[22], Mary I. O'Connor[23], Nessa E. O'Connor[24], Jeanine L. Olsen[10], Robert J. Orth[25], Bradley J. Peterson[26], Henning Reiss[27], Francesca Rossi[28], Jennifer Ruesink[29], Erik E. Sotka[30], Jonas Thormar[31], Fiona Tomas[32], Richard Unsworth[12], Erin P. Voigt[4], Matthew A. Whalen[33,34], Shelby L. Ziegler[35] and John J. Stachowicz[1]

[1]Department of Evolution and Ecology, and [2]DataLab: Data Science and Informatics, University of California, Davis, CA, USA
[3]Tennenbaum Marine Observatories Network, MarineGEO, Smithsonian Environmental Research Center, Edgewater, MD, USA
[4]Department of Biology, San Diego State University, San Diego, CA, USA
[5]Department of Environmental and Marine Biology, Åbo Akademi University, Åbo, Finland
[6]Estuary & Ocean Science Center and Department of Biology, San Francisco State University, San Francisco, CA, USA
[7]Sciences fondamentales and Québec Océan, Université du Québec à Chicoutimi, Chicoutimi, Quebec, Canada
[8]Department of Ecology, Environment and Plant Sciences (DEEP), Stockholm University, Stockholm, Sweden
[9]CCMAR, Universidade do Algarve, Faro, Portugal
[10]University of Groningen, Groningen, The Netherlands
[11]Institute of Marine Sciences, University of North Carolina at Chapel Hill, Morehead City, NC, USA
[12]Department of Biosciences, Swansea University, Swansea, UK
[13]Universidad Autónoma de Baja California, Mexicali, Baja CA, Mexico
[14]Fisheries Research and Education Agency, Hatsukaichi, Hiroshima, Japan
[15]Department of Marine and Environmental Sciences, Northeastern University, Nahant, MA, USA
[16]Department of Ichthyology and Hydrobiology, St Petersburg State University, St Petersburg, Russia
[17]Instituto de Ciencias Polares, Ambiente y Recursos Naturales, Universidad Nacional de Tierra del Fuego, Ushuaia, Tierra del Fuego, Antártida e Islas del Atlántico Sur, Argentina
[18]University of Zadar, Zadar, Croatia
[19]Department of Biological Sciences, Pusan National University, Busan, South Korea
[20]Department of Environmental Sciences, University of Virginia, Charlottesville, VA, USA
[21]Department of Marine Sciences, University of Gothenburg, Goteborg, Sweden
[22]Hokkaido University, Akkeshi, Hokkaido, Japan
[23]Biodiversity Research Centre and Department of Zoology, University of British Columbia, Vancouver, British Columbia, Canada
[24]School of Natural Sciences, Trinity College Dublin, Dublin, Republic of Ireland
[25]Virginia Institute of Marine Science, College of William and Mary, Gloucester Point, VA, USA
[26]School of Marine and Atmospheric Sciences, Stony Brook University, Stony Brook, NY, USA
[27]Nord University, Bodø, Norway
[28]Centre National de la Récherche Scientifique, ECOSEAS Laboratory, Université de Cote d'Azur, Nice, France
[29]Department of Biology, University of Washington, Seattle, WA, USA
[30]Grice Marine Laboratory, College of Charleston, Charleston, SC, USA

[31]Institute of Marine Research, His, Norway
[32]IMEDEAS (CSIC), Esporles, Islas Baleares, Spain
[33]Hakai Institute, Campbell River, British Columbia, Canada
[34]University of British Columbia, Vancouver, British Columbia, Canada
[35]Moss Landing Marine Laboratories, Moss Landing, CA, USA

CPG, 0000-0002-0896-8476; JED, 0000-0001-8595-6391;
KAH, 0000-0002-1643-1847; MRK, 0000-0002-2729-9167;
PLR, 0000-0002-0177-3537; CB, 0000-0003-2845-8331;
KEB, 0000-0003-2680-2493; MC, 0000-0002-2111-4803;
JE, 0000-0001-6936-0926; AHE, 0000-0002-9579-9606;
BKE, 0000-0003-4752-922X; FJF, 0000-0001-8253-9648;
JNG, 0000-0003-3295-6480; CMH, 0000-0002-2088-9295;
MH, 0000-0002-4677-9377; MVI, 0000-0002-8277-7387;
PJ, 0000-0002-6018-7124; CK, 0000-0003-4255-8400;
K-SL, 0000-0003-0431-1829; JL, 0000-0002-8787-1786;
P-OM, 0000-0001-8611-7848; MN, 0000-0002-5722-3585;
MIO, 0000-0001-9583-1592; NEO, 0000-0002-3133-0913;
RJO, 0000-0003-2491-7430; BJP, 0000-0001-5942-8253;
HR, 0000-0003-1393-0269; FR, 0000-0003-1928-9193; JR, 0000-0001-5691-2234;
EES, 0000-0001-5167-8549; JT, 0000-0002-7925-3822; RU, 0000-0003-0036-9724;
EPV, 0000-0003-3415-7842; MAW, 0000-0002-5262-6131;
SLZ, 0000-0001-7218-0811; JJS, 0000-0003-2735-0564

While considerable evidence exists of biogeographic patterns in the intensity of species interactions, the influence of these patterns on variation in community structure is less clear. Studying how the distributions of traits in communities vary along global gradients can inform how variation in interactions and other factors contribute to the process of community assembly. Using a model selection approach on measures of trait dispersion in crustaceans associated with eelgrass (*Zostera marina*) spanning 30° of latitude in two oceans, we found that dispersion strongly increased with increasing predation and decreasing latitude. Ocean and epiphyte load appeared as secondary predictors; Pacific communities were more overdispersed while Atlantic communities were more clustered, and increasing epiphytes were associated with increased clustering. By examining how species interactions and environmental filters influence community structure across biogeographic regions, we demonstrate how both latitudinal variation in species interactions and historical contingency shape these responses. Community trait distributions have implications for ecosystem stability and functioning, and integrating large-scale observations of environmental filters, species interactions and traits can help us predict how communities may respond to environmental change.

## 1. Introduction

Community ecology is fundamentally concerned with the assembly and maintenance of diversity across space and time. Key to this endeavour is the idea that the composition of a local community is the result of multiple ecological filters selecting species from a regional pool [1,2]. Different kinds of filters apply different kinds of selective pressures on the species pool, and because species' traits are what allow them to pass through filters, studying the distribution of traits within the community can help us understand how these filters act on the species pool as a whole. Strong environmental filters (i.e. abiotic filters *sensu* [3]) such as climate are thought to act on large spatial scales to constrain trait diversity such that species

are more alike (clustered) in traits that respond to these factors than we would expect under a purely random assembly process [2,4–6]. Biotic filters, such as competition, then act at smaller spatial scales to enhance or reduce trait diversity among species with broadly similar abiotic tolerances, depending on which traits are affected [7]. When traits related to the acquisition of distinct resources are considered, competition for these resources drives the distribution of traits to be wider than expected by chance (overdispersed) as there are multiple resource niche optima that can be occupied [4,5,8]. By contrast, competition for a single, dominant limiting resource can also act as a filter, selecting for traits related to acquiring this resource to converge around an optimal value, because species deviating from the optimum are otherwise competitively excluded. All else being equal, as richness increases, an increase in trait dispersion may point to stronger stabilizing mechanisms and limiting similarity, while a decrease in trait dispersion can suggest stronger equalizing mechanisms promoting unstable coexistence [7,9].

Despite well-known geographical patterns in the strength of both biotic interactions and environmental filters [10–13], few studies have examined the global-scale consequences of geographical variation in these filters for community trait distributions [14,15]. In particular, intense predation, competition, and mutualistic interactions at lower latitudes [12,13,16], may lead to the predominance of biotic interactions over environmental filters in structuring low-latitude communities. This may cause stronger trait clustering near the poles that shifts towards more overdispersed communities at lower latitudes. On the other hand, selection for tolerance of extreme heat conditions could also cause trait clustering at low latitudes. Finally, patterns in community structure along latitudinal gradients could be dominated by idiosyncratic and historically contingent effects of predators, prey, competitors, and mutualists that vary among biogeographic provinces [17–20]. Local abiotic factors, habitat complexity, assemblage composition and adaptation to these local factors could further obscure broader geographical patterns of community assembly [17,21], stressing the importance of assessing patterns across multiple independent species pools. For example, the effects of regional gradients in predation may be overshadowed by local increases in habitat complexity, which can decrease predation pressure [11] and increase trait dispersion as species assort into disparate microhabitat niches [22]. Understanding trait distributions and their drivers should provide insight into the likely responses of communities to environmental fluctuations or perturbations in the same way that understanding the diversity of traits within a population can inform us on its evolutionary potential [23,24].

Here, we examine geographical patterns in the trait distribution of epifaunal invertebrates living on eelgrass throughout the Northern Hemisphere to assess the extent and causes of geographical variation in the drivers of the assembly of these communities. Eelgrass (*Zostera marina*) is the world's most widespread species of temperate seagrass, a marine angiosperm found throughout the Northern Hemisphere from 30° to 67° N latitude in both the Atlantic and Pacific Oceans [25,26]. Much of the animal community in eelgrass beds is made up of invertebrate mesograzers that primarily feed on the epiphytic microalgae fouling the seagrass blades [27]. Competition for food and microhabitat space occurs among mesograzers, and can significantly affect community composition [22,28–30]. Peracarid crustaceans (amphipods, isopods and tanaids) are the most widespread, abundant and species-rich mesograzer taxon in these eelgrass beds, and they

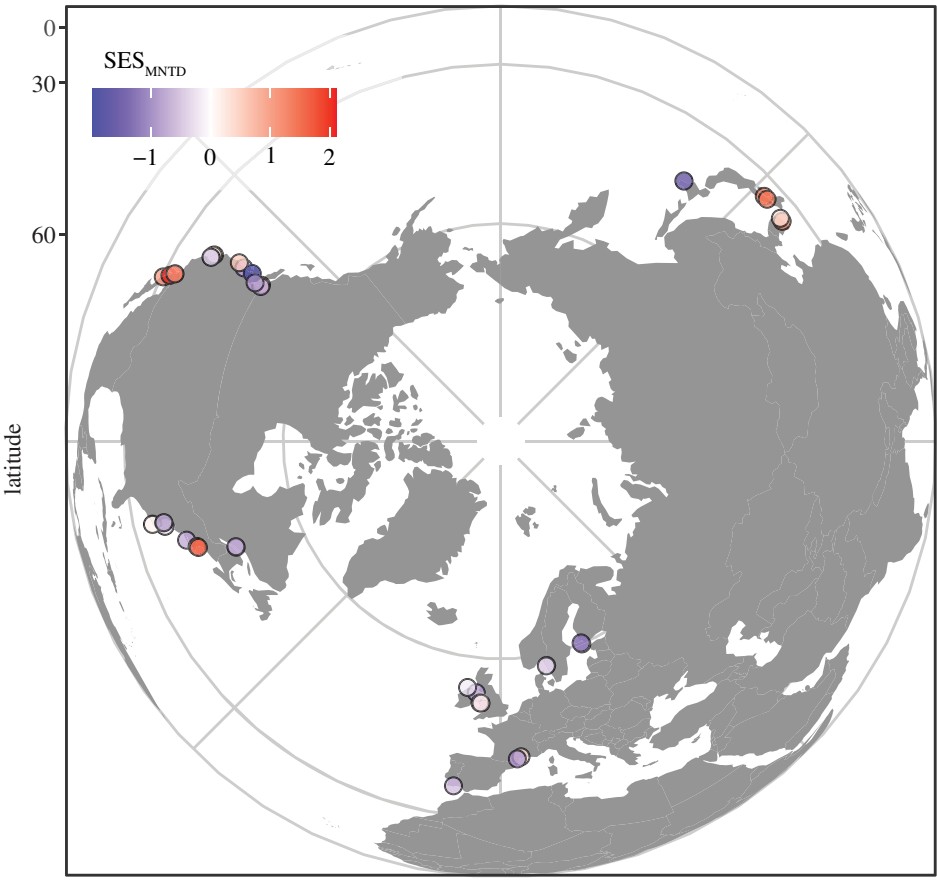

**Figure 1.** *Zostera* experimental network (ZEN) sites used in our analyses. Sites spanned 30° of latitude on the Pacific and Atlantic coasts of North America and Eurasia, including the Baltic and Mediterranean seas, covering most of the range of *Zostera marina* (eelgrass). Colours indicate trait dispersion (standard effect size, mean nearest taxon distance (SES$_{MNTD}$) calculated using the tip shuffle algorithm); positive values of SES$_{MNTD}$ indicate greater dispersion in traits than expected from a random draw from the global species pool, whereas negative values of SES$_{MNTD}$ indicate clustering in traits relative to a random draw. See the electronic supplementary material, figure S1 for more detailed information about site locations. (Online version in colour.)

experience elevated predation in low-latitude eelgrass beds [11] which could either cause clustering of communities around traits that increase resistance or tolerance to predation, or cause dispersion of communities owing to competition for enemy-free space. *Zostera marina*'s wide range across latitudes provides an opportunity to assess the role of gradients of ecological filters on global scales without the confounding influence of changing habitat type. We predicted: (i) that trait dispersion would increase with decreasing latitude as species interactions become more intense, and (ii) that abiotic filters would be strongest and result in clustering at higher latitudes and where biotic interactions are weak. While marine systems often show nonlinear variation in species diversity and interaction strength with latitude (peaking at mid-latitudes; [20,31]), our predictions are reasonable within the range of latitudes occupied by eelgrass (approx. 30–70° N). We test these predictions in separate ocean basins with largely unique fauna, allowing us to assess whether the unique histories of these zoogeographic provinces result in different patterns and drivers of trait distribution in each ocean basin [32,33].

## 2. Methods

### (a) Study design and sample collection

Between May and September 2014, we sampled 42 sites across the range of *Z. marina*, spanning 30 degrees of latitude along both coasts of Eurasia and North America (30.4° N to 60.1° N; figure 1) to characterize the biological and physical structure of eelgrass beds using standardized measurements. We implemented a hierarchical sampling design consisting of two oceans (Atlantic and Pacific), each with two coasts (east and west), each with 6–14 sites, each with 20 plots, for a total of 840 plots in 42 sites sampled as part of the *Zostera* Experimental Network (ZEN; electronic supplementary material, figure S1). Plots were 1 m$^2$ and spaced 2 m apart at each site. Along each coastline, sites were separated by 4.9 km (Virginia, USA) to 485.4 km (Washington State, USA) of water.

### (b) Assessing eelgrass habitat characteristics

We sampled eelgrass biomass by haphazardly placing and pushing a 20 cm diameter core tube 20 cm into the sediment within each plot. We gathered all shoots rooted within the core bottom area into the core tube to ensure that no shoots were cut off during sampling. We then removed the shoots from the sediment and transferred the core contents into a mesh bag. In the laboratory, we rinsed the core contents, removed fouling algae and sediment from the eelgrass tissue, and separated above- and belowground biomass by cutting the plant above the rhizome. In addition to eelgrass, we also removed all of the macroalgae from the plot. All eelgrass and macroalgal tissue was dried to a constant weight at 60°C and weighed. From five haphazardly collected eelgrass shoots per plot, we also collected 3 cm lengths of tissue from a healthy, unfouled inner leaf and processed these samples for tissue nitrogen using a CHN analyser (Thermo Fisher Scientific Inc., Waltham, MA, USA).

**Table 1.** Traits used in analyses of ZEN peracarid communities. (Sources for individual species traits are listed in electronic supplementary material, appendix S1.)

| trait | type | values | category | interpretation | citations |
|---|---|---|---|---|---|
| maximum fecundity (number of eggs) | ordered categorical | very low (0 to <18), low (18 to <31), medium (31 to <65), high (65 to <135), very high (>135) | neither | competitive ability, population resilience, population density | [38–41] |
| maximum adult length | continuous | 2–50 mm | microhabitat | susceptibility to predators, ability to occupy physical space | [38–41] |
| body shape | categorical | laterally compressed, dorsoventrally compressed, vermiform | microhabitat | ability to occupy physical space, palatability | [40] |
| living habit | categorical | free, parasite/direct commensal, tube/burrow dweller | microhabitat | degree of substrate association, substrate type, population density | [39,41] |
| motility | categorical | swimmer, crawler | microhabitat | susceptibility to predators, dispersal ability, degree of substrate association | [40,41] |
| bioturbator | binary | | microhabitat | degree of substrate association, substrate type | [41] |
| microalgae feeding | binary | | diet | dietary niche partitioning | [39,42,43] |
| macroalgae feeding | binary | | diet | dietary niche partitioning | [39,42,43] |
| seagrass feeding | binary | | diet | dietary niche partitioning | [39,42,43] |
| seagrass detritus feeding | binary | | diet | dietary niche partitioning | [39,42,43] |
| suspension feeding | binary | | diet | dietary niche partitioning | [39,42,43] |
| detritivory, deposit feeding | binary | | diet | dietary niche partitioning | [39,42,43] |
| carnivory, parasitism, scavenging | binary | | diet | dietary niche partitioning | [39,42,43] |

We quantified eelgrass habitat structure at the plot level by measuring shoot density and canopy height. We estimated shoot density by counting the number of shoots emerging within a 20 cm diameter ring placed haphazardly in the plot. In plots where density was particularly low (less than 50 shoots m$^{-2}$, about 5% of plots), we counted all of the shoots in the plot. We measured canopy height by haphazardly collecting five shoots from each plot and measuring their length from the tip of the longest leaf to the leaf sheath.

We sampled epiphyte load on the eelgrass blades by selecting four shoots from each plot and removing them from the substrate either by gently uprooting or clipping at the meristem and placing them in a plastic bag on ice for transport. In the laboratory, we scraped both sides of all the leaves with a glass slide to remove fouling material, which was then filtered, transferred to an aluminium pan, dried to a constant weight at 60°C, and weighed.

## (c) Measuring predation intensity

Predation intensity was quantified by tethering locally-collected prey ('gammarid' amphipods) in each plot for 24 h. These data and methods are reported in detail in Reynolds et al. [11]. Briefly, each individual amphipod was glued to a 10 cm piece of monofilament line 0.133 mm in diameter (Berkley Fireline™, Spirit Lake, IA, USA) tied to a transparent acrylic stake anchored in the sediment, so that it could swim freely in the water column and cling to adjacent eelgrass blades. After 24 h, we removed the stakes and scored prey as present (uneaten) or absent (eaten); partially consumed prey were considered eaten, and moulted prey were excluded from analyses. Site-level predation was calculated by averaging scores across plots.

## (d) Abiotic environmental variables

To characterize the abiotic environment experienced by epifauna across the range of eelgrass, we measured in situ temperature and salinity at each site at the time of sampling. To characterize the overall abiotic environment of each site, we also retrieved estimates of annual mean sea surface temperature (SST), photosynthetically active radiation (PAR), and surface chlorophyll a (Chl a) from the surrounding region, available in the Bio-ORACLE dataset [34]. These data were taken from monthly readings of the Aqua-MODIS and SeaWiFS satellites at a 9.6 km$^2$ spatial resolution from 2002 to 2009. We used the raster package in R v. 3.6.3 [35,36] to extract the annual mean SST, SST range, PAR and Chl a from all cells within 10 km of each site, and averaged these cell-level estimates to generate site-level predictors. Other water quality parameters, including dissolved nitrate and other nutrients, were spatially interpolated based on surface measurements in the World Ocean Database 2009 [37].

## (e) Epifaunal community composition

To sample the macrofauna associated with the eelgrass blades, we carefully placed an open-mouthed fine-mesh drawstring bag (500 μm mesh, 18 cm diameter) over a clump of shoots in the centre of the plot so that the mouth of the bag was flush with the sediment surface. We then cut the shoots where they emerged from the sediment and quickly closed the drawstring to capture the shoots and associated animals. The shoots were transferred to the laboratory on ice, rinsed and hand-inspected to dislodge the epifauna, which were then passed through a 1 mm sieve and ultimately transferred into 70% ethanol. Epifauna were then identified to the lowest possible taxonomic level (typically species). Epifaunal abundance was standardized by the aboveground biomass of the eelgrass sample from which they were collected.

We scored all peracarids (amphipods, isopods and tanaids) for a series of traits based on information available in the literature, including body size, fecundity, body shape, living habit, motility, bioturbation and diet components (table 1; electronic supplementary material, appendix S1 for the literature). Owing to a paucity of data on intraspecific trait variation for most species, literature values were assumed to be representative for all individuals in our study. For subsequent analyses, we categorized each of these traits as related to microhabitat or dietary niche; we also performed analyses with all traits ungrouped. While we acknowledge that these broad categories may overlap, we elected to sort traits into these categories because they represent two potential components of trait dispersion exhibited by peracarids in field studies and laboratory experiments [22,29]. Correlations among traits were generally weak, save for strong positive relationships between eating live seagrass tissue and macroalgae, detritivory and consuming seagrass detritus, and suspension feeding and bioturbation (electronic supplementary material, figure S2).

## (f) Characterizing community dispersion

For all the peracarid species observed in our dataset, we used the trait dataset to generate three matrices of Gower distances between species: one of all traits, one for diet traits and one for microhabitat traits using the FD package in R [44]. Using subsets of these matrices for communities at the site level (summed across 20 plots at each site, $n = 42$), we measured the trait distance between species as the mean pairwise distance (MPD) and mean nearest taxon distance (MNTD) for each set of traits [4,45]. MPD is the average of the trait distances between all pairs of species found within a given sample unit (site), while MNTD is the average minimum distance between species pairs in a site. Both are independent of species richness, but the two metrics can behave differently depending on the clustering of species in trait space within a sample [45].

To determine whether the observed species traits in each community differed from those expected by chance, we standardized MPD and MNTD against null distributions generated according to two permutation algorithms. The first, independent swap, is a semi-constrained model that randomly re-assembles the sample-by-species community matrix while maintaining the species richness of each sample and the presence/absence of each species across samples. The second, tip shuffle, is a more constrained model that directly shuffles the traits of the species in the community while maintaining richness, occurrence, and trait distances between community members, effectively moving the tip labels on a trait dendrogram. Imposing more constraints on permutation controls for patterns in the data that are not directly relevant to the question at hand, such as species richness, occurrence, or identity, ultimately reducing type I error rates [46]. Because of the relatively low overlap in species pools across the range of our study, comparing the results relative to both types of models can be informative of the importance of species identity in these types of permutations, and also facilitate comparison with other studies in which the independent swap algorithm has been used together with less constrained permutations (e.g. [22]). These permutations were each completed 999 times for each community, and null distributions of MPD and MNTD were generated based on values calculated from randomized communities.

We examined the effect of the species pool on community dispersion, using varying degrees of constraint on the matrix and trait dendrogram used to generate null distributions. To make comparisons among sites, we permuted within the global species pool (all sites) and ocean-level Atlantic and Pacific species pools. Using a global pool in our permutations is appropriate because while all species were not present in all regions,

there were no traits that were exclusive to any region (electronic supplementary material, figure S2).

Each observed value of community trait distance was then compared to the corresponding null distribution by calculating the standard effect size ($SES_{MPD}$ or $SES_{MNTD}$). A positive value of SES indicates that the observed community trait distance (as measured by MPD or MNTD) is greater than the null mean, meaning that community members are more dissimilar than expected under a random draw (overdispersion), while a negative SES indicates that trait distance is less than the null mean, meaning that community members are more similar to each other than expected under a random draw (clustering). MPD, MNTD, null distributions and SES values were calculated using the picante package in R [47].

## (g) Data analysis

Two distance metrics (MPD and MNTD), two permutation algorithms (independent swap and tip shuffle), three species pools (global, Pacific, and Atlantic), and three trait sets (all, diet, and microhabitat) totalled 36 sets of SES values. However, owing to missing diet data for some species, we were unable to calculate diet $SES_{MNTD}$ with the tip shuffle algorithm, leaving us with a total of 33 sets. For each distance metric, algorithm, species pool and trait set, SES values were used as response variables in a set of 16 linear models incorporating latitude, ocean, continental margin (east versus west), *in situ* temperature and salinity, annual mean and range of SST, total crustacean abundance and median crustacean size, epifaunal and peracarid richness, macroalgal biomass, average predation intensity, epiphyte load, Chl *a*, PAR, water column nitrate, mean leaf nitrogen content, and two axes of eelgrass habitat structure as derived from a principal component (PC) analysis incorporating shoot density, leaf sheath width and length, longest leaf length, and aboveground biomass (PC1 and 2; electronic supplementary material, figure S4) as predictor variables, as well as select interactions between them (table 2). Predictors were log-, square-root-, or arcsin-transformed where appropriate to conform to a normal distribution based on Shapiro–Wilk normality tests and visual examinations of histograms. Collinearity of predictors was accounted for using variance inflation factors (VIF) for variables in composite models using the car package in R [48]. Predictors with a VIF greater than five were removed from composite models. We also examined the effects of predictors on the SES of individual traits to understand what traits may drive the patterns we see across environmental gradients (electronic supplementary material, appendix S2).

We ranked these initial hypothesis-driven models of SES using Akaike information criterion corrected for small sample size (AICc) scores (MuMIn package; [49]), and then incorporated predictors from the three lowest-scoring models of each set into a set of composite models to examine the combined effects of multiple predictor types. We then used backwards elimination to select the lowest-scoring model from these composite models. Where two models had a ΔAICc less than 3 units, we selected the model with the fewest parameters for interpretation.

## 3. Results

Peracarid assemblages at Pacific sites had greater trait dispersion than Atlantic sites, and dispersion increased with increasing predation and decreasing latitude, though there were some differences among the two oceans that we outline below. Across our sites, we found a total of 105 species, 55 of which were found in the Atlantic, and 60 of which were found in the Pacific, with 10 species found in both oceans. There were 15 species in the northwest Pacific, 48 species in

**Table 2.** A priori models used to analyse site-level SES values. (These 16 models were separately applied to 33 sets of SES values for different trait distance metrics, permutation algorithms, species pools and trait sets, for a total of 528 models.)

| model name | predictors |
| --- | --- |
| biogeography 1 | latitude |
| biogeography 2 | latitude, continental margin, ocean |
| biogeography 3 | latitude, continental margin, latitude × continental margin |
| biogeography 4 | latitude, continental margin, ocean, latitude × continental margin |
| biogeography 5 | latitude, continental margin, ocean, latitude × continental margin, latitude × ocean |
| abiotic environment | *in situ* temperature, *in situ* salinity, mean leaf % N |
| temperature regime 1 | mean SST |
| temperature regime 2 | SST range |
| temperature regime 3 | mean SST, SST range, mean SST × SST range |
| community | log(mean standard total crustacean abundance), median crustacean size |
| total biodiversity | log(site epifaunal richness) |
| peracarid biodiversity | log(site peracarid richness) |
| habitat | PC1, PC2, log(macroalgal biomass + 1) |
| predation | arcsin(mean amphipod predation) |
| resource 1 | log(mean epiphyte load), log(mean Chl *a*) |
| resource 2 | $\sqrt{NO_2}$, mean photosynthetically active radiation |

the northeast Pacific, 36 species in the northwest Atlantic and 24 species in the northeast Atlantic (electronic supplementary material, figure S3). The patterns and predictors of trait dispersion were robust across SES metrics and permutation algorithms (electronic supplementary material, table S1 and figure S5); here, we present and interpret the results of model selection on SES$_{MNTD}$ calculated using the tip shuffle algorithm, with exceptions presented where relevant.

## (a) Dispersion of traits by ocean basin

Of the set of all traits examined, communities at Atlantic sites were on average clustered (SES < 0) relative to the global null, particularly for body size and living habit (electronic supplementary material, figure A2–2)—species clustered around a mean body size of 14.09 mm (47.5% smaller than the mean Pacific body size), and most were free-living. Communities at Pacific sites were overdispersed (SES > 0) on average relative to the global null (figure 2; electronic supplementary material, tables S1 and S2). This pattern held for both metrics and null models but was significant only for SES$_{MPD}$ (SES$_{MPD}$ independent swap $t_{38.097} = 2.43$, $p = 0.020$; SES$_{MPD}$ tip shuffle $t_{38.242} = 2.31$, $p = 0.027$; two-sample $t$-tests). Within the global pool, the separate calculations of SES using microhabitat and feeding traits showed a similar pattern; for microhabitat traits, Pacific communities were more overdispersed and Atlantic communities more clustered (SES$_{MNTD}$ tip shuffle $t_{35.654} = 3.64$, $p = 0.00086$; figure 2).

## (b) Correlates of among-site variation in trait dispersion

Predation intensity, latitude, epiphyte load and ocean basin (within the global species pool) were the strongest and most consistent predictors of SES across all species pools and all trait sets (electronic supplementary material, table S1 and figure S5). *In situ* temperature, bed characteristics, epifaunal richness, continental margin, nitrate and salinity

also appeared occasionally (less than 30% of models) across the best models of SES. Mean annual sea surface temperature, epifaunal richness, salinity, nitrate, *in situ* temperature and crustacean abundance also varied significantly with latitude (electronic supplementary material, figure S8).

In all of the best models, peracarid communities at sites with higher predation intensity had more overdispersed traits, whereas those with less intense predation had more clustered traits relative to a random draw from the species pool (figure 3a; electronic supplementary material, table S1, figure S5a–c). Predation (removal of amphipod baits) varied from 20% in Quebec to 100% in Sweden, San Francisco Bay, Ireland, Korea and British Columbia; the average predation rate was significantly greater in the Pacific than in the Atlantic Ocean (electronic supplementary material, table S3 and figures S7, S8), but this did not translate to a difference in the effect of predation on dispersion across the two basins when permuting within the global pool ($p = 0.48$; figure 3a). Across the three species pools, the predation effect was stronger on average when permuting within the Pacific than the Atlantic or global pools, (electronic supplementary material, figure S5a), and strongest in models of the dispersion of all traits together (electronic supplementary material, figure S5b).

As predicted, trait dispersion decreased with increasing latitude in the best models (global species pool, microhabitat traits); communities became more clustered at higher latitude, while communities towards the equatorward edge of *Z. marina*'s range were more overdispersed (figure 3b; electronic supplementary material, figure S5d–f). These latitude effects were stronger in the Pacific Ocean than in the Atlantic ($F_{1,38} = 7.95$, $p = 0.0076$; figure 3b) although they did not appear in the top models when permuting within the Pacific species pool (electronic supplementary material, figure S5d); the best model including latitude was 1.3 AICc units better than the top model, but it was not selected as the top model because of the small difference in AICc score and

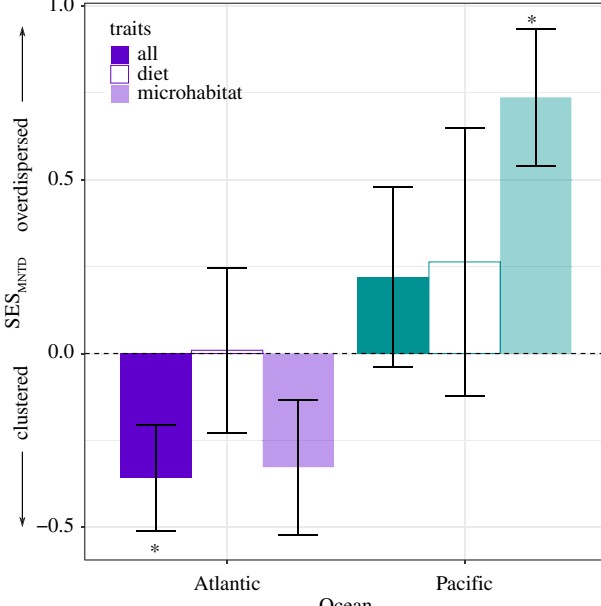

**Figure 2.** Trait dispersion (SES$_{MNTD}$) in eelgrass-associated peracarid crustacean communities across trait sets. In general, communities at sites in the Pacific Ocean were more overdispersed, while communities at Atlantic sites were less dispersed than expected. The dashed horizontal line represents an SES$_{MNTD}$ value of 0, indicating random assembly. Asterisks indicate means significantly different from zero (two-tailed one-sample *t*-tests; see the electronic supplementary material, table S2); error bars represent standard errors. The figure shows SES$_{MNTD}$ calculated according to the tip shuffle permutation algorithm; results were comparable across permutation algorithms and SES values. (Online version in colour.)

greater number of parameters. Like predation, the latitude effect was strongest in models including all traits together (electronic supplementary material, figure S5e).

Communities were more clustered (more negative SES) at sites with high epiphyte loads, but this effect was most obvious in the Atlantic species pool when only microhabitat traits were considered (figure 3c; electronic supplementary material, figure S5 g-h). There was rarely an effect of epiphyte load on SES when using other species pools (electronic supplementary material, figure S5 g and table S1) and never for diet traits (electronic supplementary material, figure S5h).

## 4. Discussion

Using a global dataset of eelgrass-associated peracarid crustaceans, we found a strong increase in community trait dispersion with decreasing latitude and increasing predation (figure 3a,b). Latitudinal clines in different ecological filters have been well characterized in a wide variety of systems [10,11,13], particularly temperature and the strength of species interactions [10,12,13], both of which decrease at high latitudes. Stronger biotic interactions, in particular stabilizing interactions (*sensu* [9]), at lower latitudes may select for an overdispersed community [4,7,8], while stronger abiotic filters (or relatively weaker biotic filters) at either end of range (e.g. cold at the poleward edge or hot at the equatorward edge) could select for a clustered community [3–5]. We found similar total numbers of species in the two oceans (electronic supplementary material, figure S3) given similar sampling effort, and all traits were found in both oceans, so the differences

we observe among oceans are not simply the result of different diversities in the underlying species pool.

Several lines of evidence point to the relatively greater effect of biotic interactions over temperature in structuring our communities. First, temperature rarely appeared as a significant factor in our best models (figure 3d; electronic supplementary material, table S1). Second, latitudinal clines in dispersion were more dependent on ocean basin than continental margins, which differ significantly in their temperature gradients (western side of oceans are warmer at an equivalent latitude; figure 3b; [11]). Third, predation in this system decreases with latitude, as it does in many others [11–13]. Fourth, we observed greater dispersion in living habit, motility and macroalgae consumption at lower latitudes (electronic supplementary material, figure A2–1b-d), all of which can be reasonably linked to stabilizing competition for food or enemy-free space. Finally, for some traits (body size, fecundity), we would expect clustering at both ends of a thermal gradient, but around different optima: large-bodied and highly fecund peracarids at cool sites, and small-bodied peracarids that produce fewer eggs at warm sites [38,50]. However, in ectotherms like peracarids, decreases in temperature at higher latitudes are less likely to be strong drivers of community structure than increases in temperature at lower latitudes as a result of asymmetrical performance curves [51,52]. While we saw that high-latitude sites tended to have species with high fecundity (65 to <135 eggs per brood; part of a general trend for clustered sites to have high or very high fecundity; electronic supplementary material, figure A2–1a), we saw no similar trend towards clustering at low latitudes around low fecundity values or any other traits.

The decline in trait dispersion with latitude was significantly greater in the Pacific than the Atlantic. This difference in latitudinal clines and trait dispersion more generally between the two ocean basins (figures 2 and 3b) may be in part owing to differences in these assemblages' biogeographic and evolutionary histories [18]. First, glaciation in the north Atlantic during the last Ice Age means that many of the areas in which eelgrass now occurs would have been colonized after glaciers retreated [53,54], leaving less time for *in situ* adaptation and specialization that might lead to increased trait dispersion [5]. Similarly, given *Z. marina*'s origin in the Pacific and more recent Pleistocene expansion into the Atlantic [54], we might also generally expect Atlantic species to have colonized eelgrass from other Atlantic-native habitats, perhaps predisposing them to be less overdispersed in their traits as they cluster around a single mean. Consistent with this, we found that species in Atlantic sites were clustered around a smaller mean body size, which may be selected for by the denser eelgrass habitat in the Atlantic (electronic supplementary material, figures S4, A2–2a; [55]. Finally, gastropod relative abundance increases with latitude, and gastropods are a more abundant and speciose component of the epifaunal community in the north Atlantic than in the Pacific (Gross *et al.* 2021, unpublished). Competition with gastropods for epiphytes or other shared resources may push the peracarids there into a more constrained area of trait space, leading to the clustering we observed.

The precise impacts of these and other historical factors are difficult to quantify but may be further investigated with analyses of phylogenetic dispersion or more detailed studies of

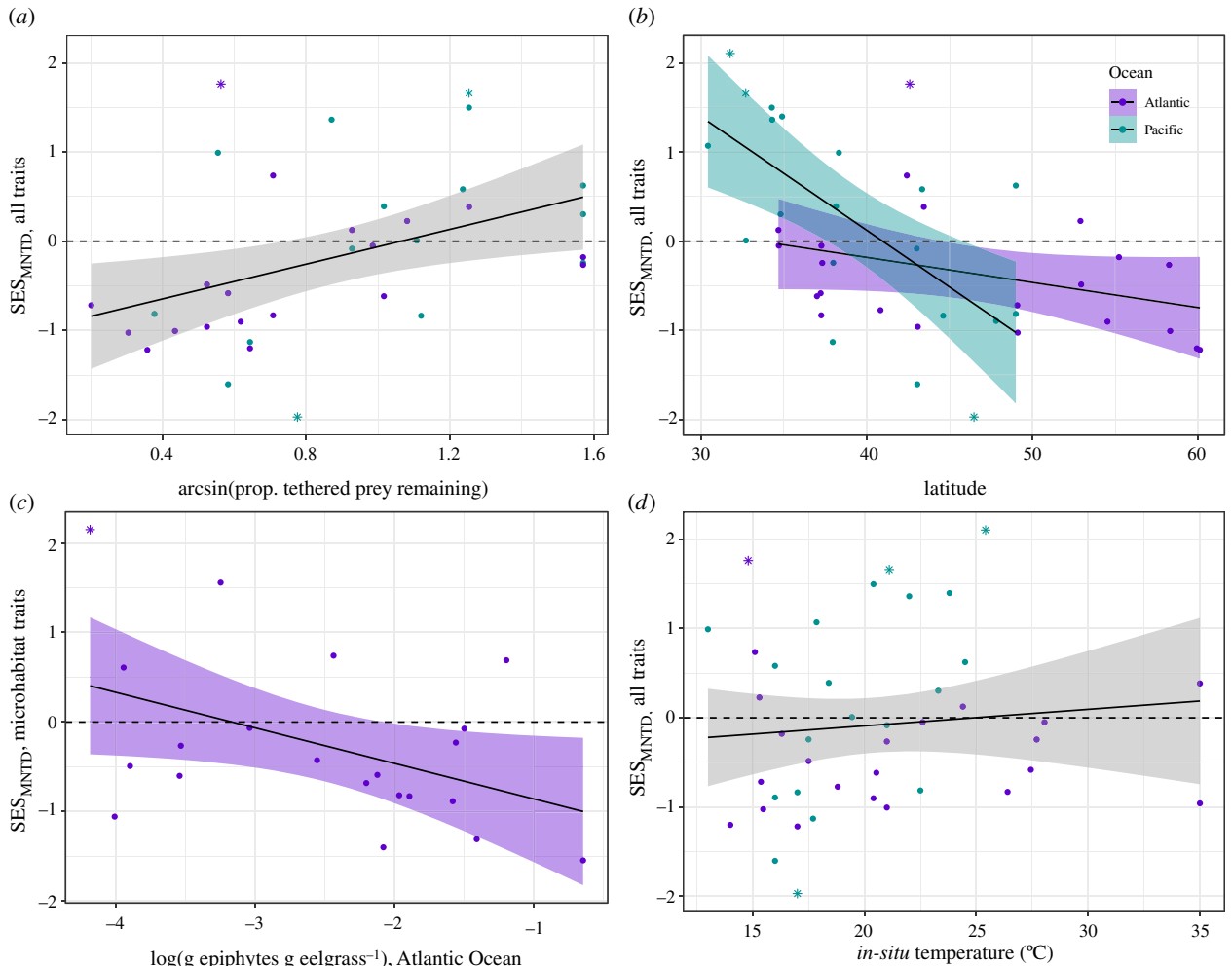

**Figure 3.** The effects of predation (*a*), latitude (*b*), epiphyte load (*c*) and *in situ* temperature (*d*) on trait dispersion (SES$_{MNTD}$ using the tip shuffle algorithm) in univariate analyses. In all of the best models of dispersion, sites with higher predation intensity had more overdispersed communities, while those with lower predation intensity had more clustered communities (*a*; $R^2 = 0.15$, $p = 0.012$). In the best models that had a non-zero latitude effect, sites at lower latitudes had more overdispersed communities, while those at higher latitudes had more clustered communities. This effect was stronger in the Pacific than the Atlantic species pool (*b*; $R^2 = 0.36$, interaction $p = 0.0076$). In the best models with a non-zero epiphyte effect, sites where eelgrass had lower epiphyte density had more overdispersed communities, while sites with more heavily fouled blades had clustered communities (*c*; plot shows SES$_{MNTD}$ for microhabitat traits in the Atlantic species pool; $R^2 = 0.15$, $p = 0.046$). *In situ* temperature appeared only sporadically across permutations and dispersion metrics, and was not significant for total trait dispersion ($R^2 = 0.0094$, $p = 0.54$). The dashed horizontal line represents an SES value of 0, indicating random assembly; sites in bold italics are those for which SES is significantly different from 0 at $\alpha = 0.05$. See the electronic supplementary material, figure S1 for an explanation of site codes. (Online version in colour.)

trait distributions in the regional species pool [15,56]. However, we currently lack a phylogeny of peracarids with sufficient resolution and taxon sampling with which to evaluate underlying differences in phylogenetic diversity between the two ocean basins. We do note that richness of species, genera and families did not vary substantially between the ocean basins (electronic supplementary material, figure S3).

One of the most striking results of our study was the positive effect of predation intensity on community dispersion among sites that was consistent in both oceans (figure 3*a*); peracarid species were more dissimilar in their traits than expected by chance in sites with high predation intensity. This effect appeared across trait sets, species pools, dispersion metrics and methods (electronic supplementary material, table S1), although we rarely saw this signal at the level of individual traits (electronic supplementary material, table A2–1, figure A2–3). Changes in predator community structure, predation intensity, or both could lead to an increase in competition for predator-free space, an ecological selective filter that may result in overdispersion, particularly with respect to microhabitat and predator avoidance traits [22].

Herbivorous arthropods in both marine and terrestrial systems are known to select their microhabitat niches based largely on their effectiveness as shelter from predators rather than the availability or quality of food [57–59]. Consequently, competition for enemy-free space can be an important factor structuring communities. Alternatively, predation could affect trait dispersion by reducing competition [30,60], but we would expect this to lead to an increase in dispersion from strongly clustered (SES < 0) to random communities (SES = 0) as stabilizing competition lessened, rather than the observed shift from clustered to overdispersed (SES > 0, figure 3*a*; electronic supplementary material, figure S5b).

Latitudinal patterns of species interactions are now broadly appreciated [10–13,16,20], but rarely are these results explicitly connected to variation in the structure of communities. By examining both how species interactions and environmental drivers vary within a single habitat type across a broad geographical gradient, we demonstrate an important role for latitudinal variation in species interactions in driving patterns of community assembly. Diversity in important traits can increase the completeness with which

epiphytes are removed, leading to increased seagrass growth [61], an effect that is strongest in the presence of predators [62]. More generally, trait clustering and dispersion have implications for redundancy, stability, and ecosystem functioning [5,23,63]. For instance, communities may be less resilient to environmental change if they are clustered by environmental filters [23,24]. Clustering that occurs as a result of equalizing mechanisms (*sensu* [9]) can weaken the relationship between diversity and ecosystem functioning, or certain ecosystem functions may be enhanced in communities with overdispersed effect traits, especially if diversity-function relationships arise through complementarity [2,63]. Thus, historical contingency and broad-scale ecological drivers may play an important role in constraining not only the assembly of local communities, but the resulting trait diversity can affect the functioning of the entire ecosystem. This approach, if applied broadly, offers the potential for developing a predictive understanding of how entire communities respond to environmental change.

**Data accessibility.** The data are provided in the electronic supplementary material [64].

**Authors' contributions.** All authors contributed to data curation, investigation, methodology, review, and editing. C.P.G., M.R.K., and J.J.S. conceptualized the manuscript; C.P.G. conducted formal analyses; and C.P.G. and J.J.S. wrote the original draft. Funding was secured by J.E.D, K.A.H., and J.J.S.; the project was administered by J.E.D. and P.L.R., and supervised by J.J.S. All authors gave final approval for publication and agreed to be held accountable for the work performed therein.

**Competing interests.** We declare we have no competing interests.

**Funding.** This research was funded by National Science Foundation grants to J.E.D., J.J.S. and K.A.H. (NSF-OCE 1336206, OCE 1336905, and OCE 1336741). C.B. was funded by the Åbo Akademi University Foundation. This manuscript was prepared as a chapter for C.P.G.'s doctoral dissertation.

**Acknowledgements.** We thank the many laboratory and field assistants that participated in this research and whose contributions of time and effort were invaluable for making this project happen. The manuscript was improved with comments from SP Lawler, ED Sanford, SY Strauss and two anonymous referees.

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
