## [Peer Review File · Proceedings of the Royal Society B: Biological Sciences]

Review History

RSPB-2021-1762.R0 (Original submission)

Review form: Reviewer 1

Recommendation

Accept with minor revision (please list in comments)

Scientific importance: Is the manuscript an original and important contribution to its field?

Excellent

General interest: Is the paper of sufficient general interest?

Excellent

Quality of the paper: Is the overall quality of the paper suitable?

Good

Is the length of the paper justified?

Yes

Should the paper be seen by a specialist statistical reviewer?

No

Do you have any concerns about statistical analyses in this paper? If so, please specify them explicitly in your report.

No

It is a condition of publication that authors make their supporting data, code and materials available - either as supplementary material or hosted in an external repository. Please rate, if applicable, the supporting data on the following criteria.

Is it accessible?

Yes

Is it clear?

Yes

Is it adequate?

Yes

Do you have any ethical concerns with this paper?

No

Comments to the Author

This study presents a large-scale test of the relative importance of biotic and abiotic factors in predicting the among species variation in traits within a community. Using the mobile crustaceans associated with eelgrass across 42 sites in the Pacific and Atlantic, the authors found that trait dispersion increased at low latitudes where predation intensity was highest. Given the recent interest in how species interactions vary with latitude in both terrestrial and marine communities, this is a very welcome addition to the literature as it links that variation to trait assembly within communities.

My main suggestions for improving the manuscript relate to 1) a clearer organisation of the predictor variables used to ensure better justification for their inclusion in the models, 2) revised presentation of the results in the figures to include more of the biological results (trait dispersion vs a wider range of the abiotic and biotic predictors) and less of the robustness of those findings across different analytical approaches, and 3) revision of the discussion to clarify expectations about differences in speciation among basins. Other comments are below, by line number.

143. The abstract could be improved by a sentence explaining why trait dispersion is of primary interest (i.e., the link between community structure in the first sentence and the desire to understand variance among species in trait values as presented in the first paragraph of the introduction).

227. While there is an expectation that competition will be more intense at low latitudes (line 196), I am not sure it could be assumed for the peracarid communities inhabiting eelgrass. There remains relatively little experimental work to demonstrate the strength of interspecific competition in communities of mobile epifauna. Graham Edgar has some nice experiments to show that they community was food limited, but if there are more recent works that quantify

competition it would be good to include them here.

Similarly, Whalen et al. (2020 PNAS) including many of the coauthors here had non-linear relationships between predation on artificial baits and latitude in seagrass beds. If competition and predation in this system cannot be assumed to covary in a predictable fashion with latitude, this first aim can be reframed as testing how trait dispersion varies with latitude rather than varies with known variation in those predictors.

295. I appreciate that intraspecific variation in traits will not be available for most of these species, but how body size is treated would seem very important. Most taxa will have skewed distributions with few individuals reaching the maximum adult length used here. Given that predation is frequently size selective, geographical variation in predation could result in shifts in size distributions within species but not variation in the species present across sites.

350. There is a long list of potential predictor variables that include abiotic predictors and characteristics of the both the epifaunal community (e.g., abundance) and the eelgrass habitat structure. The introduction ended with the general aim of relating trait dispersion to latitude and abiotic filters (which I assumed to be things like temperature, salinity), but didn't provide the background as to why community metrics like crustacean abundance or richness would be potential predictor variables. Similarly, it would be good to have more background on how the eelgrass habitat traits were considered – clearly an important potential covariate but is there an expectation of how variation in shoot and leaf traits would relate to epifaunal trait dispersion. These could be the local factors as presented in line 204, or potentially vary on large geographic scales also. Overall, it would be good to group these predictors into the logical framework presented in the introduction (paragraph starting line 191).

372-377. Seems odd to have much of the first paragraph in the result dedicated to the reasons why the tip shuffle algorithm is presented. I suggest presenting the main biological findings and then state that the results were robust and did not differ greatly with the alternative permutation methods (supported by the supplementary figures).

Fig. 2. Given that the permutation methods differed little, this figure could be simplified to just include the tip shuffle method as mostly presented elsewhere in the results. Then you have a bit more space to write out full names of the response variable (e.g., avoiding the SES_MNTD abbreviations where it is not immediately obvious that high values represent overdispersion – could annotate the figure to show positive values are over-dispersed and negative clustered). I like to aim for a figure where the reader can get the main results without too much reference to the figure legend.

Fig. 2. The legend for ocean basin could be put within the box for panel b allowing all three figures to have the same aspect ratio. The units for a and b are somewhat abstracted – can they have units that could be more easily interpreted by the reader (eg mg epiphytes/cm² leaf area). I can understand the desire to put present the three best predictors, but I also think it is informative to visualise some of the other relationships that test the key predictions (eg trait dispersion vs temperature). This is the key figure that presents the overall test of how trait dispersion varied with the range of biotic and abiotic variables presenting in the introduction as likely to be important.

This is a fabulous dataset and I don't feel like the choice of results presented here vs the supplementary figures, or aesthetics of the data visualisation, do justice to the important ideas being tested.

434. The Reynolds study presented here supports the prediction that predation will vary predictably with latitude for this system (see comment for line 227 above).

454. Relate these comments to that actual observed species richness in the different regions, as presented in Fig S3 – conflicts with the statement at the start of the next paragraph with similar

richness in both basins.

461, Fig S4. There are many supplementary figures, but I am left wanting to visualise the distribution of the predictor variables with latitude and across the ocean basins. This figure goes some way to doing that for the habitat traits, but would like to see that also for other predictors used.

472. Link to comments in the previous paragraph about time for speciation in the different biogeographic regions.

Review form: Reviewer 2

Recommendation

Major revision is needed (please make suggestions in comments)

Scientific importance: Is the manuscript an original and important contribution to its field?

Excellent

General interest: Is the paper of sufficient general interest?

Excellent

Quality of the paper: Is the overall quality of the paper suitable?

Good

Is the length of the paper justified?

Yes

Should the paper be seen by a specialist statistical reviewer?

No

Do you have any concerns about statistical analyses in this paper? If so, please specify them explicitly in your report.

No

It is a condition of publication that authors make their supporting data, code and materials available - either as supplementary material or hosted in an external repository. Please rate, if applicable, the supporting data on the following criteria.

Is it accessible?

Yes

Is it clear?

Yes

Is it adequate?

Yes

Do you have any ethical concerns with this paper?

No

Comments to the Author

This study connects latitudinal patterns of species interactions, specifically the intensity of predation, with variation in the structure of epifaunal crustacean communities and the

distribution of species traits. Impressive global sampling efforts produced a community-level dataset that includes species' presence/absence, abundance and traits, predation intensity, habitat characteristics, abiotic environmental variables, and community-composition. To my knowledge, not many datasets like this exist for any system on such a large scale. This large-scale framework adds robustness to the result that increasing predation intensity increases trait dispersion. However, there are some questions left unanswered regarding the ocean-specific nature of predation intensity. For example, was the range of predation intensity comparable between oceans? And does trait dispersion increase with increasing predation in both the Atlantic and Pacific? It's not impossible to discover the answers to these questions with the information provided, but they are important pieces of information that should be directly presented.

Since we know very little about how predation influences the assembly (and evolution) of species within communities at the macroscale, partly because most studies of biotic interactions focus on the effects of competition, the fact that this study focuses on predation makes it an important contribution that will be of general interest to a wide readership.

Overall, I found this study incredibly interesting. The authors have done a nice job introducing the topics and highlighting the implications of their work. The overall quality of the writing, clarity and presentation is good. However, I do have a major concern that should be addressed, and I believe there are multiple ways the authors may do so. The study is from a decidedly community assembly point of view. Yet, much of the discussion focuses on historical contingency and differences in biogeography between the Atlantic and Pacific Oceans (Lines 447-471). The issue is that the patterns of trait dispersion and clustering observed could be the result of differences in the phylogenetic relatedness of taxa in each community/ocean basin – differences in evolutionary history. If traits evolved via Brownian motion and taxa in the Pacific were more distantly related than those in the Atlantic (which is not unlikely given the recency of eelgrass habitat expansion into the Atlantic) then traits would also be more dispersed in the Pacific and more clustered in the Atlantic, without any other effects of predation intensity [see Gerhold, P, Cahill, JF, Winter, M, Bartish, IV, & Prinzing, A. (2015). Phylogenetic patterns are not proxies of community assembly mechanisms (they are far better). *Functional Ecology*, 29(5), 600–614]. This is one reason why it is important to present the relationship between predation intensity and trait dispersion for each ocean basin. Ideally, comparisons of phylogenetic diversity for each community and ocean basin would shed more light on the relative effects of predation intensity. For example, if the Atlantic and Pacific have comparable phylogenetic diversity, this could be addressed in the discussion like the way similar species richness was addressed (Lines 472-479). However, it's unclear whether a species-level phylogeny is available for peracarids. If a phylogeny is available, it could also be used to simulate trait evolution under Brownian motion to create null distributions for comparison. This would reveal the level of trait dispersion expected given the evolutionary relatedness of taxa in a community (under Brownian motion) and the remaining trait dispersion could be related to other factors. If a phylogeny is not available, the authors should directly address this limitation and the possibility for differences in evolutionary history to drive the patterns observed.

A less-major concern pertains to the authors' prediction that abiotic filters would be strongest and result in clustering at higher latitudes where biotic interactions are weak. My concern is whether the authors have shown that clustering at higher latitudes is actually due to abiotic filters being stronger there? Or whether they have shown that the **relative** effects of abiotic factors are stronger at higher latitudes because biotic interactions are weaker there (as implied in Lines 429-430)? This is probably a semantic issue, but I think it's worth clarifying. I'm not sure how one would go about measuring the absolute strength of an abiotic filter, but in my mind that wasn't the goal of the study anyways.

Specific comments:

Line 209: understanding THE diversity of traits...

Line 212: geographic patterns in THE trait distribution...

Lines 226-227: I think the first prediction could be stated more simply to help the reader link it to the background information provided earlier in the discussion. I had a hard time understanding exactly what this prediction was and why it makes sense. Maybe, something like 'we predict traits will be more dispersed at lower latitudes where the strength of competition and predation are strongest'. Or 'we predict trait dispersion will increase with increasing the strength of competition and predation, which is highest at lower latitudes'. Even the reverse prediction would be easier to understand, 'we predict that trait dispersion would increase with decreasing latitude as the strength of competition and predation increase'. It could be further simplified by removing competition, especially since the results are focused on predation intensity.

Lines 264-271: How exactly were the scores of eaten and uneaten prey used to quantify predation intensity? Was the average taken across all plots?

Lines 294-302: How is a trait only related to microhabitat niche but not dietary niche? For example, body size and many other traits would influence both. In Table 1 maximum adult length is interpreted in the context of feeding rates but categorized as microhabitat. How might such distinct categorizations impact the results and the conclusions drawn? This should be acknowledged/discussed somewhere. Moreover, how is a trait related to neither microhabitat niche nor dietary niche? And how was this determined? It seems like there would need to be concrete evidence to conclude that a particular trait is not at all related to dietary niche, as even traits like maximum fecundity would influence nutrient requirements.

Please make sure all R packages used are detailed and cited alongside the appropriate methods.

Lines 398-403: Since predation intensity had a strong effect on trait dispersion, more details should be provided in the main text to summarize the range of predation intensity observed overall and within each ocean basin. A supplemental table summarizing the range of values for all the site data by ocean basin would also provide useful context.

Line 401: Are there one too many uses of the word average in this sentence?

Line 449: And differences in their evolutionary histories.

Lines 473-474: But they could be due to differences in the relatedness of species in the underlying species pool.

Lines 494-496: I enjoyed the discussion in this paragraph, but I'm wondering whether the result that predation also caused overdispersion of diet traits could be discussed in the context of additional theory? For example, how does this result support/not support the idea that predation can lower competition by regulating prey populations. See:

Pianka, E. R. Latitudinal gradients in species diversity: a review of concepts. *The American Naturalist* 100, 33-46 (1966).

Line 729, Figure 3: There should be a '' after similar patterns. Also, is it the case that dispersion increases with increasing predation for both ocean basins? The contribution of each ocean basin to this overall pattern would be clearer if the site labels were coloured as in panel b.

Line 746, Figure S2: It might be helpful to include a definition of Amphi- in the figure legend for those who are unfamiliar with the prefix.

Decision letter (RSPB-2021-1762.R0)

12-Oct-2021

Dear Mr Gross:

Your manuscript has now been peer reviewed and the reviews have been assessed by an Associate Editor. The reviewers' comments (not including confidential comments to the Editor) and the comments from the Associate Editor are included at the end of this email for your reference. As you will see, the reviewers and the Editors have raised some concerns with your manuscript and we would like to invite you to revise your manuscript to address them.

Research ethics:

Use of animals and field studies:

It is a condition of publication that you make available the data and research materials supporting the results in the article. Please see our Data Sharing Policies (<https://royalsociety.org/journals/authors/author-guidelines/#data>). Datasets should be deposited in an appropriate publicly available repository and details of the associated accession number, link or DOI to the datasets must be included in the Data Accessibility section of the

article (<https://royalsociety.org/journals/ethics-policies/data-sharing-mining/>). Reference(s) to datasets should also be included in the reference list of the article with DOIs (where available).

If you wish to submit your data to Dryad (<http://datadryad.org/>) and have not already done so you can submit your data via this link [http://datadryad.org/submit?journalID=RSPB&manu=\(Document not available\)](http://datadryad.org/submit?journalID=RSPB&manu=(Document%20not%20available)), which will take you to your unique entry in the Dryad repository.

Please submit a copy of your revised paper within three weeks. If we do not hear from you within this time your manuscript will be rejected. If you are unable to meet this deadline please let us know as soon as possible, as we may be able to grant a short extension.

Best wishes,
Dr Daniel Costa
mailto: proceedingsb@royalsociety.org

Associate Editor
Board Member: 1
Comments to Author:

This manuscript presents interesting results relating to latitudinal patterns of species interactions (predation), with variation in the structure of communities and the distribution of species traits. This manuscript is clearly written, and the results are novel. The manuscript has been reviewed by two reviewers. Both found aspects of this paper interesting, but Reviewer 2 pointed out a number of ways in which the manuscript can improve. In particular, Reviewer 2 asks for further details regarding the ocean-specific nature of predation intensity. In addition, Reviewer 2 provides a comprehensive summary of several refinements related to abiotic filter predictions and flags an issue with the patterns of trait dispersion and clustering observed that should be addressed in order to improve the manuscript. Further, Reviewer 1 highlights several ways in which the ms could improve in relation to how the work is organised and presented. These revisions will ultimately improve the manuscript.

Reviewer(s)¹ Comments to Author:

Referee: 1

Comments to the Author(s)

This study presents a large-scale test of the relative importance of biotic and abiotic factors in predicting the among species variation in traits within a community. Using the mobile crustaceans associated with eelgrass across 42 sites in the Pacific and Atlantic, the authors found that trait dispersion increased at low latitudes where predation intensity was highest. Given the recent interest in how species interactions vary with latitude in both terrestrial and marine communities, this is a very welcome addition to the literature as it links that variation to trait assembly within communities.

My main suggestions for improving the manuscript relate to 1) a clearer organisation of the predictor variables used to ensure better justification for their inclusion in the models, 2) revised presentation of the results in the figures to include more of the biological results (trait dispersion vs a wider range of the abiotic and biotic predictors) and less of the robustness of those findings across different analytical approaches, and 3) revision of the discussion to clarify expectations about differences in speciation among basins. Other comments are below, by line number.

143. The abstract could be improved by a sentence explaining why trait dispersion is of primary interest (i.e., the link between community structure in the first sentence and the desire to understand variance among species in trait values as presented in the first paragraph of the introduction).

227. While there is an expectation that competition will be more intense at low latitudes (line 196), I am not sure it could be assumed for the peracarid communities inhabiting eelgrass. There remains relatively little experimental work to demonstrate the strength of interspecific competition in communities of mobile epifauna. Graham Edgar has some nice experiments to show that they community was food limited, but if there are more recent works that quantify competition it would be good to include them here.

Similarly, Whalen et al. (2020 PNAS) including many of the coauthors here had non-linear relationships between predation on artificial baits and latitude in seagrass beds. If competition and predation in this system cannot be assumed to covary in a predictable fashion with latitude, this first aim can be reframed as testing how trait dispersion varies with latitude rather than varies with known variation in those predictors.

295. I appreciate that intraspecific variation in traits will not be available for most of these species, but how body size is treated would seem very important. Most taxa will have skewed distributions with few individuals reaching the maximum adult length used here. Given that predation is frequently size selective, geographical variation in predation could result in shifts in size distributions within species but not variation in the species present across sites.

350. There is a long list of potential predictor variables that include abiotic predictors and characteristics of the both the epifaunal community (e.g., abundance) and the eelgrass habitat structure. The introduction ended with the general aim of relating trait dispersion to latitude and abiotic filters (which I assumed to be things like temperature, salinity), but didn't provide the background as to why community metrics like crustacean abundance or richness would be potential predictor variables. Similarly, it would be good to have more background on how the eelgrass habitat traits were considered – clearly an important potential covariate but is there an expectation of how variation in shoot and leaf traits would relate to epifaunal trait dispersion. These could be the local factors as presented in line 204, or potentially vary on large geographic scales also. Overall, it would be good to group these predictors into the logical framework presented in the introduction (paragraph starting line 191).

372-377. Seems odd to have much of the first paragraph in the result dedicated to the reasons why the tip shuffle algorithm is presented. I suggest presenting the main biological findings and

then state that the results were robust and did not differ greatly with the alternative permutation methods (supported by the supplementary figures).

Fig. 2. Given that the permutation methods differed little, this figure could be simplified to just include the tip shuffle method as mostly presented elsewhere in the results. Then you have a bit more space to write out full names of the response variable (e.g., avoiding the SES_MNTD abbreviations where it is not immediately obvious that high values represent overdispersion – could annotate the figure to show positive values are over-dispersed and negative clustered). I like to aim for a figure where the reader can get the main results without too much reference to the figure legend.

Fig. 2. The legend for ocean basin could be put within the box for panel b allowing all three figures to have the same aspect ratio. The units for a and b are somewhat abstracted – can they have units that could be more easily interpreted by the reader (eg mg epiphytes/cm² leaf area). I can understand the desire to put present the three best predictors, but I also think it is informative to visualise some of the other relationships that test the key predictions (eg trait dispersion vs temperature). This is the key figure that presents the overall test of how trait dispersion varied with the range of biotic and abiotic variables presenting in the introduction as likely to be important.

This is a fabulous dataset and I don't feel like the choice of results presented here vs the supplementary figures, or aesthetics of the data visualisation, do justice to the important ideas being tested.

434. The Reynolds study presented here supports the prediction that predation will vary predictably with latitude for this system (see comment for line 227 above).

454. Relate these comments to that actual observed species richness in the different regions, as presented in Fig S3 – conflicts with the statement at the start of the next paragraph with similar richness in both basins.

461, Fig S4. There are many supplementary figures, but I am left wanting to visualise the distribution of the predictor variables with latitude and across the ocean basins. This figure goes some way to doing that for the habitat traits, but would like to see that also for other predictors used.

472. Link to comments in the previous paragraph about time for speciation in the different biogeographic regions.

Referee: 2

Comments to the Author(s)

This study connects latitudinal patterns of species interactions, specifically the intensity of predation, with variation in the structure of epifaunal crustacean communities and the distribution of species traits. Impressive global sampling efforts produced a community-level dataset that includes species' presence/absence, abundance and traits, predation intensity, habitat characteristics, abiotic environmental variables, and community-composition. To my knowledge, not many datasets like this exist for any system on such a large scale. This large-scale framework adds robustness to the result that increasing predation intensity increases trait dispersion. However, there are some questions left unanswered regarding the ocean-specific nature of predation intensity. For example, was the range of predation intensity comparable between oceans? And does trait dispersion increase with increasing predation in both the Atlantic and Pacific? It's not impossible to discover the answers to these questions with the information provided, but they are important pieces of information that should be directly presented.

Since we know very little about how predation influences the assembly (and evolution) of species within communities at the macroscale, partly because most studies of biotic interactions focus on

the effects of competition, the fact that this study focuses on predation makes it an important contribution that will be of general interest to a wide readership.

Overall, I found this study incredibly interesting. The authors have done a nice job introducing the topics and highlighting the implications of their work. The overall quality of the writing, clarity and presentation is good. However, I do have a major concern that should be addressed, and I believe there are multiple ways the authors may do so. The study is from a decidedly community assembly point of view. Yet, much of the discussion focuses on historical contingency and differences in biogeography between the Atlantic and Pacific Oceans (Lines 447-471). The issue is that the patterns of trait dispersion and clustering observed could be the result of differences in the phylogenetic relatedness of taxa in each community/ocean basin – differences in evolutionary history. If traits evolved via Brownian motion and taxa in the Pacific were more distantly related than those in the Atlantic (which is not unlikely given the recency of eelgrass habitat expansion into the Atlantic) then traits would also be more dispersed in the Pacific and more clustered in the Atlantic, without any other effects of predation intensity [see Gerhold, P, Cahill, JF, Winter, M, Bartish, IV, & Prinzing, A. (2015). Phylogenetic patterns are not proxies of community assembly mechanisms (they are far better). *Functional Ecology*, 29(5), 600–614]. This is one reason why it is important to present the relationship between predation intensity and trait dispersion for each ocean basin. Ideally, comparisons of phylogenetic diversity for each community and ocean basin would shed more light on the relative effects of predation intensity. For example, if the Atlantic and Pacific have comparable phylogenetic diversity, this could be addressed in the discussion like the way similar species richness was addressed (Lines 472-479). However, it's unclear whether a species-level phylogeny is available for peracarids. If a phylogeny is available, it could also be used to simulate trait evolution under Brownian motion to create null distributions for comparison. This would reveal the level of trait dispersion expected given the evolutionary relatedness of taxa in a community (under Brownian motion) and the remaining trait dispersion could be related to other factors. If a phylogeny is not available, the authors should directly address this limitation and the possibility for differences in evolutionary history to drive the patterns observed.

A less-major concern pertains to the authors' prediction that abiotic filters would be strongest and result in clustering at higher latitudes where biotic interactions are weak. My concern is whether the authors have shown that clustering at higher latitudes is actually due to abiotic filters being stronger there? Or whether they have shown that the relative effects of abiotic factors are stronger at higher latitudes because biotic interactions are weaker there (as implied in Lines 429-430)? This is probably a semantic issue, but I think it's worth clarifying. I'm not sure how one would go about measuring the absolute strength of an abiotic filter, but in my mind that wasn't the goal of the study anyways.

Specific comments:

Line 209: understanding THE diversity of traits...

Line 212: geographic patterns in THE trait distribution...

Lines 226-227: I think the first prediction could be stated more simply to help the reader link it to the background information provided earlier in the discussion. I had a hard time understanding exactly what this prediction was and why it makes sense. Maybe, something like 'we predict traits will be more dispersed at lower latitudes where the strength of competition and predation are strongest'. Or 'we predict trait dispersion will increase with increasing the strength of competition and predation, which is highest at lower latitudes'. Even the reverse prediction would be easier to understand, 'we predict that trait dispersion would increase with decreasing latitude as the strength of competition and predation increase'. It could be further simplified by removing competition, especially since the results are focused on predation intensity.

Lines 264-271: How exactly were the scores of eaten and uneaten prey used to quantify predation intensity? Was the average taken across all plots?

Lines 294-302: How is a trait only related to microhabitat niche but not dietary niche? For example, body size and many other traits would influence both. In Table 1 maximum adult length is interpreted in the context of feeding rates but categorized as microhabitat. How might such distinct categorizations impact the results and the conclusions drawn? This should be acknowledged/discussed somewhere. Moreover, how is a trait related to neither microhabitat niche nor dietary niche? And how was this determined? It seems like there would need to be concrete evidence to conclude that a particular trait is not at all related to dietary niche, as even traits like maximum fecundity would influence nutrient requirements.

Please make sure all R packages used are detailed and cited alongside the appropriate methods.

Lines 398-403: Since predation intensity had a strong effect on trait dispersion, more details should be provided in the main text to summarize the range of predation intensity observed overall and within each ocean basin. A supplemental table summarizing the range of values for all the site data by ocean basin would also provide useful context.

Line 401: Are there one too many uses of the word average in this sentence?

Line 449: And differences in their evolutionary histories.

Lines 473-474: But they could be due to differences in the relatedness of species in the underlying species pool.

Lines 494-496: I enjoyed the discussion in this paragraph, but I'm wondering whether the result that predation also caused overdispersion of diet traits could be discussed in the context of additional theory? For example, how does this result support/not support the idea that predation can lower competition by regulating prey populations. See:

Pianka, E. R. Latitudinal gradients in species diversity: a review of concepts. *The American Naturalist* 100, 33–46 (1966).

Line 729, Figure 3: There should be a ' ' after similar patterns. Also, is it the case that dispersion increases with increasing predation for both ocean basins? The contribution of each ocean basin to this overall pattern would be clearer if the site labels were coloured as in panel b.

Line 746, Figure S2: It might be helpful to include a definition of Amphi- in the figure legend for those who are unfamiliar with the prefix.

Author's Response to Decision Letter for (RSPB-2021-1762.R0)

See Appendix A.

RSPB-2021-1762.R1 (Revision)

Review form: Reviewer 1

Recommendation

Accept as is

Scientific importance: Is the manuscript an original and important contribution to its field?

Excellent

General interest: Is the paper of sufficient general interest?

Excellent

Quality of the paper: Is the overall quality of the paper suitable?

Excellent

Is the length of the paper justified?

Yes

Should the paper be seen by a specialist statistical reviewer?

No

Do you have any concerns about statistical analyses in this paper? If so, please specify them explicitly in your report.

No

It is a condition of publication that authors make their supporting data, code and materials available - either as supplementary material or hosted in an external repository. Please rate, if applicable, the supporting data on the following criteria.

Is it accessible?

Yes

Is it clear?

Yes

Is it adequate?

Yes

Do you have any ethical concerns with this paper?

No

Comments to the Author

The authors have done an excellent job of responding to reviewers' concerns. I have no further comments with each of my initial issues handled by appropriate edits to the text, revisions of figures to improve presentation of results or written explanations as to why no changes were made (in the case of the issue about predation vs mean prey sizes).

Please note that I have published in the past with a few of the authors on this manuscript (Duffy, Stachowicz, Sokta, Hughes, Whalen & Reynolds) and have a current grant application pending with Hughes. I had no involvement in this study.

Decision letter (RSPB-2021-1762.R1)

14-Jan-2022

Dear Mr Gross

I am pleased to inform you that your manuscript entitled "The biogeography of community assembly: latitude and predation drive variation in community trait distribution in a guild of epifaunal crustaceans" has been accepted for publication in Proceedings B.

Data Accessibility section

Open Access

You are invited to opt for Open Access, making your freely available to all as soon as it is ready for publication under a CCBY licence. Our article processing charge for Open Access is £1700. Corresponding authors from member institutions (<http://royalsocietypublishing.org/site/librarians/allmembers.xhtml>) receive a 25% discount to these charges. For more information please visit <http://royalsocietypublishing.org/open-access>.

Paper charges

Sincerely,

Dr Daniel Costa

Appendix A

28 October 2021

Dear Dr. Costa:

We thank you, the Associate Editor, and Reviewers for constructive feedback on our manuscript. We have made the recommended changes and feel that the resulting manuscript is much improved. Below we offer point-by-point responses to the comments of the associate editor and reviewers (our responses are in italics) and attach a file of the revised manuscript using tracked changes to indicate revisions, as suggested.

Please let us know if you have any further requests from us and we look forward to hearing about your decision on the revised manuscript.

Collin Gross and Jay Stachowicz (for the authors)

Associate Editor

Board Member: 1

Comments to Author:

This manuscript presents interesting results relating to latitudinal patterns of species interactions (predation), with variation in the structure of communities and the distribution of species traits. This manuscript is clearly written, and the results are novel. The manuscript has been reviewed by two reviewers. Both found aspects of this paper interesting, but Reviewer 2 pointed out a number of ways in which the manuscript can improve. In particular, Reviewer 2 provides a comprehensive summary of several refinements related to abiotic filter predictions and flags an issue with the patterns of trait dispersion and clustering observed that should be addressed in order to improve the manuscript. Further, Reviewer 1 highlights several ways in which the ms could improve in relation to how the work is organised and presented. These revisions will ultimately improve the manuscript.

We thank the editor for this summary and are pleased that the editor sees novelty in the manuscript. We address each of these points made by the reviewers below.

Referee: 1

Comments to the Author(s)

This study presents a large-scale test of the relative importance of biotic and abiotic factors in predicting the among species variation in traits within a community. Using the mobile crustaceans associated with eelgrass across 42 sites in the Pacific and Atlantic, the authors found that trait dispersion increased at low latitudes where predation intensity was highest. Given the recent interest in how species interactions vary with latitude in both terrestrial and marine communities, this is a very welcome addition to the literature as it links that variation to trait assembly within communities.

My main suggestions for improving the manuscript relate to

1) a clearer organization of the predictor variables used to ensure better justification for their inclusion in the models,

We address this comment at the comment for line 350 below.

2) revised presentation of the results in the figures to include more of the biological results (trait dispersion vs a wider range of the abiotic and biotic predictors) and less of the robustness of those findings across different analytical approaches, and

We address this comment at the comment for lines 372-377 and 461 below.

3) revision of the discussion to clarify expectations about differences in speciation among basins.

We address this comment at the comment for line 472 below.

Other comments are below, by line number.

143. The abstract could be improved by a sentence explaining why trait dispersion is of primary interest (i.e., the link between community structure in the first sentence and the desire to understand variance among species in trait values as presented in the first paragraph of the introduction).

*We have added the following sentence to address this (lines 142-143 in the final manuscript):
“Studying how the distributions of traits in communities vary along global gradients can inform how variation in interactions and other factors contribute to the process of community assembly.”*

227. While there is an expectation that competition will be more intense at low latitudes (line 196), I am not sure it could be assumed for peracarid communities inhabiting eelgrass. There remains relatively little experimental work to demonstrate the strength of interspecific competition in communities of mobile epifauna. Graham Edgar has some nice experiments to show that the community was food limited, but if there are more recent works that quantify competition it would be good to include them here.

We have added several references that demonstrate the importance of competition in mesograzers communities (lines 229-231).

Similarly, Whalen et al. (2020 PNAS) including many of the coauthors here had non-linear relationships between predation on artificial baits and latitude in seagrass beds. I competition and predation in this system cannot be assumed to covary in a predictable fashion with latitude, this first aim can be reframed as testing how trait dispersion varies with latitude rather than varies with known variation in those predictors.

We cite Whalen et al. here, both for their examination of predator species pools and overall predation rates across latitude – that paper found a negative effect of latitude on predation rates in temperate and subtropical seagrass beds (peaking around 30°N, the southern limit of the range of eelgrass). We’ve added a sentence to clarify this at the end of our introduction (lines 247-250), and also cited Reynolds et al. 2018 to support the pattern of monotonically decreasing predation with latitude in eelgrass beds using live prey (line 233).

295. I appreciate that intraspecific variation in traits will not be available for most of these species, but how body size is treated would seem very important. Most taxa will have skewed distributions with few individuals reaching the maximum adult length used here. Given that predation is frequently size-selective, geographical variation in predation could result in shifts in size distributions within species but not variation in the species present across sites.

While we acknowledge that intraspecific trait variation could be important, we focus on maximum size because our traits generally represent “potential” traits of a species rather than “realized” traits – for example, we do not know the diet of a species at a particular site, but we use literature data to describe potential overlap with other taxa. Regarding predation it is not immediately obvious to us what the effect of predation on mean size would be. Fish in eelgrass beds are usually juveniles and so very large or very small prey might avoid being eaten by them. We do have available coarse size class data for the individuals counted for the surveys. Based on this data, there is no predictable shift in mean size within species or other taxonomic groups as a function of latitude (see figure below).

Figure 1. Observed variation in body size (coarsely measured in a series of 7 sieves: 8 mm, 5.6 mm, 2.8 mm, 2 mm, 1.4 mm, 1 mm, 0.71 mm, and 0.5 mm) across latitudes. Each point represents a species at each site it was observed. Panels a, c, e, and g show the maximum observed sieve size for each species at each site; panels b, d, f, and h show the most commonly observed sieve size for each species at each site. Panels a and b show all species across sites; c and d show amphitoid amphipods, e and f show idoteid isopods, and g and h show caprellid amphipods, three taxa that were both common across sites and exhibited a wide range in body sizes.

350. There is a long list of potential predictor variables that include abiotic predictors and characteristics of both the epifaunal community (e.g. abundance) and the eelgrass habitat structure. The introduction ended with the general aim of relating trait dispersion to latitude and

abiotic filters (which I assumed to be things like temperature, salinity), but didn't provide the background as to why community metrics like crustacean abundance or richness would be potential predictor variables. Similarly, it would be good to have more background on how the eelgrass habitat traits were considered – clearly an important potential covariate but is there an expectation of how variation in shoot and leaf traits would relate to epifaunal trait dispersion. These could be the local factors as presented in line 204, or potentially vary on large geographic scales also. Overall, it would be good to group these predictors into the logical framework presented in the introduction (paragraph starting with line 191).

We add specific ideas and predictions for how some of these will affect clustering and dispersion (lines 192-195, 214-217), but to avoid increasing length by too much we did not specifically offer predictions for every factor. For example, crustacean abundance could affect dispersion if increased density led to increased strength of competitive interactions.

372-377. Seems odd to have much of the first paragraph in the results dedicated to the reasons why the tip shuffle algorithm is presented. I suggest presenting the main biological findings and then state that the results were robust and did not differ greatly with the alternative permutation methods (supported by the supplementary figures).

We agree and we have replaced the first paragraph of the results to deal more directly with the main biological findings (lines 427-436).

Fig. 2. Given that the permutation methods differed little, this figure could be simplified to just include the tip shuffle method as mostly presented elsewhere in the results. Then you have a bit more space to write out full names of the response variable (e.g. avoiding the SES_MNTD abbreviations where it is not immediately obvious that high values represent overdispersion – could annotate the figure to show positive values are over-dispersed and negative clustered). I like to aim for a figure where the reader can get the main results without too much reference to the figure legend.

We agree. Figure 2 has been simplified to show only panel D (SES_{MNTD} with tip shuffle), and the y-axis has been annotated to show which values indicate clustering and which indicate overdispersion.

Fig. 3. The legend for ocean basin could be put within the box for panel b allowing all three figures to have the same aspect ratio. The units for a and b are somewhat abstracted – can they have units that could be more easily interpreted by the reader (eg mg epiphytes/cm² leaf area). I can understand the desire to put present the three best predictors, but I also think it is informative to visualise some of the other relationships that test the key predictions (eg trait dispersion vs. temperature). This is the key figure that presents the overall test of how trait dispersion varied with the range of biotic and abiotic variables presenting in the introduction as likely to be important.

This is a fabulous dataset and I don't feel like the choice of results presented here vs the supplementary figures, or aesthetics of the data visualisation, do justice to the important ideas being tested.

We thank the reviewer for this constructive and encouraging feedback. The legend has been put in the box for panel b, the x-axes have been altered to explicitly state the variable being visualized, and an additional panel has been added for in-situ temperature.

434. The Reynolds study presented here supports the prediction that predation will vary predictably with latitude for this system (see comment for line 227 above).

We have now cited other studies here that support the prediction that predation varies predictably with latitude in other systems (line 460), and cited Reynolds et al. in the Introduction (line 233).

454. Relate these comments to that actual observed species richness in the different regions, as presented in Fig S3 – conflicts with the statement at the start of the next paragraph with similar richness in both basins.

Here we've changed our wording to better reflect what we mean – while it's true that Atlantic taxa may have had less time to speciate (not necessarily supported by the numbers of species we observed), an increased number of species wouldn't necessarily mean greater trait dispersion (lines 479).

461, Fig S4. There are many supplementary figures, but I am left wanting to visualise the distribution of the predictor variables with latitude and across the ocean basins. This figure goes some way to doing that for the habitat traits, but would like to see that also for other predictors used.

We've consolidated supplementary figures 5-7 into a single 8-panel figure, and included additional supplementary figures showing how predation rate, epiphyte load, epifaunal richness, temperature, salinity nitrate, and crustacean abundance vary by site across ocean basins and latitudes (Figures S5-8). These data are also the subject of another paper in development that will focus more explicitly on the latitudinal patterns in what are our "predictor" variables with the goal of describing geographic variation in habitat.

472. Link to comments in the previous paragraph about time for speciation in the different biogeographic regions.

This paragraph has been reduced and modified from the first version (lines 440-454).

Referee: 2

Comments to the Author(s)

This study connects latitudinal patterns of species interactions, specifically the intensity of predation, with variation in the structure of epifaunal crustacean communities and the distribution of species traits. Impressive global sampling efforts produced a community-level dataset that includes species' presence/absence, abundance and traits, predation intensity, habitat characteristics, abiotic environmental variables, and community-composition. To my knowledge, not many datasets like this exist for any system on such a large scale. This large-scale framework adds robustness to the result that increasing predation intensity increases trait dispersion.

However, there are some questions left unanswered regarding the ocean-specific nature of predation intensity. For example, was the range of predation intensity comparable between oceans? And does trait dispersion increase with increasing predation in both the Atlantic and Pacific? It's not impossible to discover the answers to these questions with the information provided, but they are important pieces of information that should be directly presented.

Thank you for these comments. We now explicitly make the points requested on lines 420-423 and Table S3.

Since we know very little about how predation influences the assembly (and evolution) of species within communities at the macroscale, partly because most studies of biotic interactions focus on the effects of competition, the fact that this study focuses on predation makes it an important contribution that will be of general interest to a wide readership.

Overall, I found this study incredibly interesting. The authors have done a nice job introducing the topics and highlighting the implications of their work. The overall quality of the writing, clarity and presentation is good. However, I do have a major concern that should be addressed, and I believe there are multiple ways the authors may do so. The study is from a decidedly community assembly point of view. Yet, much of the discussion focuses on historical contingency and differences in biogeography between the Atlantic and Pacific Oceans (Lines 447-471). The issue is that the patterns of trait dispersion and clustering observed could be the result of differences in the phylogenetic relatedness of taxa in each community/ocean basin – differences in evolutionary history. If traits evolved via Brownian motion and taxa in the Pacific were more distantly related than those in the Atlantic (which is not unlikely given the recency of eelgrass habitat expansion into the Atlantic) then traits would also be more dispersed in the Pacific and more clustered in the Atlantic, without any other effects of predation intensity [see Gerhold, P, Cahill, JF, Winter, M, Bartish, IV, & Prinzing, A. (2015). Phylogenetic patterns are not proxies of community assembly mechanisms (they are far better). *Functional Ecology*, 29(5), 600-614]. This is one reason why it is important to present the relationship between predation intensity and trait dispersion for each ocean basin. Ideally, comparisons of phylogenetic diversity for each community and ocean basin would shed more light on the relative effects of predation intensity. For example, if the Atlantic and Pacific have comparable phylogenetic diversity, this could be addressed in the discussion like the way similar species richness was addressed (Lines 472-479). However, it's unclear whether a species-level phylogeny is available for peracarids. If a phylogeny is available, it could also be used to simulate trait evolution under Brownian motion to create null distributions for comparison. This would reveal the level of trait dispersion expected given the evolutionary relatedness of taxa in a community (under Brownian motion) and the remaining trait dispersion could be related to other factors. If a phylogeny is not available, the authors should directly address this limitation and the possibility for differences in evolutionary history to drive the patterns observed.

While we lack a species-level phylogeny of the taxa in this study, we recognize the value of including parallel analyses of phylogenetic dispersion. We've added some discussion of this as it relates both to the general patterns of trait dispersion observed in the Pacific vs. Atlantic, and as potentially underlying the differences in latitude effect between the two oceans. One thing worth noting is that most of the species that use eelgrass, especially in the Atlantic, are non-obligate and likely colonized eelgrass from other habitats when it invaded the Atlantic ~ 3 million years ago. Thus, there is no expectation of phylogenetic clustering among Atlantic taxa relative to the Pacific and in fact there is no substantial difference in the number of genera or families between basins (lines 496-497). Nonetheless, a goal of ours is to add a phylogenetic component to the analysis in the future, but the phylogeny needed to do this is still far enough off that we cannot conduct this analysis at this point.

We've also explicitly stated that there was not a significant difference in the predation rate effect between the two oceans (lines 421-423) – this is an important point that we thank the reviewer for requesting.

A less-major concern pertains to the author's prediction that abiotic filters would be strongest and result in clustering at higher latitudes where biotic interactions are weak. My concern is whether the authors have shown that clustering at higher latitudes is actually due to abiotic filters being stronger there? Or whether they have shown that the **relative** effects of abiotic factors are stronger at higher latitudes because biotic interactions are weaker there (as implied in Lines 429-430)? This is probably a semantic issue, but I think it's worth clarifying. I'm not sure how one would go about measuring the absolute strength of an abiotic filter, but in my mind that wasn't the goal of the study anyways.

This is also a good point and we have added some text to make clear that this could be due to weakening biotic or strengthening abiotic factors. We have a reason why we think it is the former and not the latter and explain this a bit more and acknowledge both are possible (lines 448-449).

Specific comments:

Line 209: understanding THE diversity of traits...

Done.

Line 212: geographic patterns in THE trait distribution...

Done.

Line 226-227: I think that the first prediction could be stated more simply to help the reader link it to the background information provided earlier in the discussion. I had a hard time understanding exactly what this prediction was and why it makes sense. Maybe, something like 'we predict traits will be more dispersed at lower latitudes where the strength of competition and predation are strongest'. Or 'we predict trait dispersion will increase with increasing the strength of competition and predation, which is highest at lower latitudes'. Even the reverse prediction would be easier to understand, 'we predict that trait dispersion would increase with decreasing latitude as the strength of competition and predation increase'. It would be further simplified by removing competition, especially since the results are focused on predation intensity.

Sentence has been edited for clarity, revised sentence reads "...trait dispersion would increase with decreasing latitude as species interactions become more intense" (lines 235-236).

Lines 264-271: How exactly were the scores of eaten and uneaten prey used to quantify predation intensity? Was the average taken across all plots?

We've added a sentence clarifying this (lines 283-284).

Lines 294-302: How is a trait only related to microhabitat niche but not dietary niche? For example, body size and many other traits would influence both. In Table 1 maximum adult length is interpreted in the context of feeding rates but categorized as microhabitat. How might such distinct categorizations impact the results and the conclusions drawn? This should be acknowledged/discussed somewhere. Moreover, how is a trait related to neither microhabitat niche nor dietary niche? And how was this determined? It seems like there would need to be concrete evidence to conclude that a particular trait is not at all related to dietary niche, as even traits like maximum fecundity would influence nutrient requirements.

The reviewer raises an excellent point that the categories by which we group traits are arbitrary and that overlap exists across these categories. We have elected to keep our traits organized into these discrete categories because they represent two potential components of trait dispersion exhibited by peracarids in field studies and laboratory experiments (Best et al. 2013, Best & Stachowicz 2014, Lürig et al. 2016). To account for the overlap among trait categories, we include the results of our analyses for all traits together, and largely focus our discussion around these. Other groupings of traits are certainly possible, but to avoid adding further permutations to our analyses we did not attempt this.

Please make sure all R packages used are detailed and cited alongside the appropriate methods.
Done.

Lines 398-403: Since predation intensity had a strong effect on trait dispersion, more details should be provided in the main text to summarize the range of predation intensity observed overall and within each basin. A supplemental table summarizing the range of values for all the site data by ocean basin would also provide useful context.

We've provided a summary of the range of predation rates across sites and the average predation rate in each ocean basin in the Results section of the text (lines 418-423) as well as in a supplemental table (Table S3).

Line 401: Are there one too many uses of the word average in this sentence?
Edited for clarity (lines 423-425).

Line 449: And differences in their evolutionary histories.
Added (line 475-477).

Lines 473-474: But they could be due to differences in the relatedness of species in the underlying species pool.

As mentioned above, we do not have a phylogeny for these species yet, but there do not seem to be major differences in diversity among oceans at any level of taxonomic organization.

Lines 494-496: I enjoyed the discussion in this paragraph, but I'm wondering whether the result that predation also caused overdispersion of diet traits could be discussed in the context of additional theory? For example, how does this result support/not support the idea that predation can lower competition by regulating prey populations. See: Pianka, E. R. Latitudinal gradients in species diversity: a review of concepts. *The American Naturalist* 100, 33-46 (1966).

We've presented a consideration of predation-mediated decrease in competition as an explanation for the simultaneous increase in diet and microhabitat trait dispersion within the context of our data (line 512-516).

Line 729, Figure 3: There should be a ‘.’ after similar patterns. Also, is it the case that dispersion increases with increasing predation for both ocean basins? The contribution of each ocean basin to this overall pattern would be clearer if the site labels were coloured as in panel b.

Site labels have been colored in each panel according to ocean basin. There was not a significant interaction between ocean and predation rate on dispersion within the global species pool, and we've clarified this in our results.

Line 746, Figure S2: It might be helpful to include a definition of Amphi- in the figure legend for those who are unfamiliar with the prefix.

Done.